# miRNAs and lncRNAs in *Echinococcus* and Echinococcosis

**DOI:** 10.3390/ijms21030730

**Published:** 2020-01-22

**Authors:** Zhi He, Taiming Yan, Ya Yuan, Deying Yang, Guangyou Yang

**Affiliations:** 1College of Animal Science and Technology, Sichuan Agricultural University, Chengdu 611130, Sichuan, China; zhihe@sicau.edu.cn (Z.H.); yantaiming@sicau.edu.cn (T.Y.); yuanya145@hotmail.com (Y.Y.); 2Farm Animal Genetic Resources Exploration and Innovation Key Laboratory of Sichuan Province, Sichuan Agricultural University, Chengdu 611130, Sichuan, China; 3College of Veterinary Medicine, Sichuan Agricultural University, Chengdu 611130, Sichuan, China; Guangyou1963@aliyun.com

**Keywords:** non-coding RNA, miRNA, lncRNA, *Echinococcus*, echinococcosis

## Abstract

Echinococcosis are considered to be potentially lethal zoonotic diseases that cause serious damage to hosts. The metacestode of *Echinococcus multilocularis* and *E. granulosus* can result in causing the alveolar and cystic echinococcoses, respectively. Recent studies have shown that non-coding RNAs are widely expressed in *Echinococcus* spp. and hosts. In this review, the two main types of non-coding RNAs—long non-coding RNAs (lncRNAs) and microRNAs (miRNAs)—and the wide-scale involvement of these molecules in these parasites and their hosts were discussed. The expression pattern of miRNAs in *Echinococcus* spp. is species- and developmental stage-specific. Furthermore, common miRNAs were detected in three *Echinococcus* spp. and their intermediate hosts. Here, we primarily focus on recent insights from transcriptome studies, the expression patterns of miRNAs and lncRNAs, and miRNA-related databases and techniques that are used to investigate miRNAs in *Echinococcus* and echinococcosis. This review provides new avenues for screening therapeutic and diagnostic markers.

## 1. Introduction

The metacestode form of *Echinococcus* spp. cestode parasites can result in the echinococcosis in the visceral organs (such as the liver and lung) of intermediate hosts. *E. granulosus* sensu lato (s.l.) and *E. multilocularis* are the two of the most common and researched harmful parasites [1]. Worldwide zoonoses are of great concern to public health; echinococcosis occurs worldwide and yet the amplitude of its severity is thought to be overlooked by the World Health Organization [1] and indeed occurs worldwide [2]. Cystic echinococcosis (CE) is typically the result of accidental ingestion of *E. granulosus* eggs and initiating detrimental effects to the liver and lungs [3] (Figure 1A). Mature adult *E. granulosus* tapeworms were found in the small intestine of the definitive host, a carnivore, are secreted in faecal matter releasing segments, or proglottids, comprised of large quantities of eggs, contaminating proximate vegetation and water sources. The larva then penetrate the intestinal wall of the intermediate hosts (such as sheep) and migrate through the circulation to various throughout the hosts body, in most cases, the liver and lungs. Whole cysts of *E. granulosus* are comprised of a cyst wall (CW) and hydatid cyst fluid. The brood capsules, protoscoleces, and free daughter cysts in hydatid cyst fluid are collectively referred to as hydatid sand [4]. Cystic echinococcosis can be characterised by the long-term growth of hydatid cysts in mammalian intermediate hosts and humans, imposing a substantial burden of disease and treatment. Alveolar echinococcosis is principally indicated by a tumour-like growth comprised of metacestodes in the rodents liver and, occasionally, human liver—caused by *E. multilocularis* [5,6] (Figure 1B). Adult worms of *E. multilocularis* living in the intestine of the host (e.g., foxes) are capable of producing vast quantities of eggs. Following the ingestion of contaminated liquids or food by small rodents, the eggs develop into metacestodes that predominantly encyst in the liver [7]. Subsequent ingestion of contaminated liquids or foods by small rodents develops metacestodes derived from the eggs, which produce large numbers of protoscoleces [7]. Though it is less frequent, occasionally humans ingest the eggs through contaminated food and water sources, mainly resulting in alveolar echinococcosis of the liver [8]. The current governmental measures implemented, although effective, cannot adequately eradicated alveolar echinococcosis and reliably control widespread *E. multilocularis* infection, particularly in some pastoral areas and mountainous in China [9,10]. Alveolar echinococcosis is considered to be one of the most dangerous parasite zoonoses to threaten humans in the world [7]. The prognosis and medical costs of patients largely depend on how early the infection of the parasite species is detected, the immune status of the individual, and the level of access to medical services [8]. Another *Echinococcus* species, *E. canadensis*, belongs to the *E. granulosus* sensu lato (s.l.) complex [11]. The metacestode of *E. canadensis* is characterised by the presence of a combination of key structures, such as an unilocular fluid-filled cyst, being identifiable by its cyst wall. Consisting of an inner germinal layer and an outer acellular laminated layer, surrounded by host-originated adventitial layer [12]. The small immature worms and protoscoleces are developed from the germinal layer of the cyst wall [12]. The intermediate host of *E. canadensis* is predominantly ungulates, however, occasionally, it can be humans [12].

The major portion of a given genome is transcribed as non-coding RNAs [13]. The genomes of *Echinococcus* spp. have similar characteristics; approximately 10–14% of the genome constitutes protein-coding genes, while the remaining genes are transcribed as non-coding RNAs [14,15]. Non-coding RNAs can be classified into two types, according to their molecular weight. Short RNAs, such as microRNAs (miRNAs, 18~24 nucleotides (nt)), are <200 nt in length. Long non-coding RNAs (lncRNAs) are longer than 200 nt and they contain long non-coding RNAs (lncRNAs) and circular RNAs (circRNAs) [13,16] (Figure 1C). Previous studies have reported that non-coding RNAs play important roles in physiological and pathological processes in eukaryotes [17]. MiRNAs and lncRNAs are widely expressed in *Echinococcus* spp., such as *E. granulosus*, *E. multilocularis,* and *E. canadensis* [18,19]. MiRNAs are small, yet powerful, regulatory RNAs that are implicated in the post-transcriptional regulation of almost all the cellular signaling pathways in both the animals and plants [20]. The seed sequence of mature miRNAs, which is 2~7 nt long, has been identified as the most essential region for the recognition of target mRNAs, lncRNAs, and circRNAs through base pair complementarity [21,22] (Figure 2). An increasing number of miRNAs that are widely expressed in *Echinococcus* spp. have been identified [23]. The parasite’s autologous proteins and miRNAs transferred to host cells, where they undergo adaptations to produce parasitic forms, when parasitic helminths reach the gut [6,24,25]. Thus, miRNAs may be key regulatory factors that are involved in the helminth-host interaction [26]. The transcription of intergenic miRNAs depends on the favouring of the original genes over expanding targets in different cell types, tissues, developmental stages, and is one of the main biofunctional pathways in parasites. These observations suggest that miRNAs are important in *Echinococcus* spp. and echinococcosis and, thus, may serve as diagnostic or treatment targets.

LncRNAs are longer than 200 nt and they have lower levels of sequence conservation and expression than protein-coding genes do with an mRNA-like structure, and some have a poly(A) tail [27]. The mechanism of action of lncRNAs is very complex [28] and it is still not entirely understood within the scientific community. Therefore, to date, the lncRNA profile of *Echinococcus* spp. has not yet been reported. However, lncRNA patterns have been described in several the intermediate hosts, such as sheep [29], goats [30], cattle [31], swine [32], horses [33], and humans [34], and the definitive host, canines [35]. LncRNAs were identified as an important regulatory factor that is crucial in regulating immunological stress during cystic echinococcosis, in an experimental mouse model of cystic echinococcosis [19]. In addition, lncRNAs have also been identified as important in the biological functions of other parasites (e.g., *Trichomonas vaginalis* [36] and *Toxoplasma* [37]) and in host immunity. Thus, we speculate that lncRNAs may be expressed in *Echinococcus* spp. and perform large-scale biological functions in both parasites and hosts.

## 2. Widespread Expression of miRNAs in *Echinococcus* spp. According to Transcriptome Analysis

Large miRNAs, being identified in *E. granulosus*, *E. multilocularis*, and *E. canadensis* (G7), have been found to feature certain characteristics of genes in expression regulation, such as tissue and developmental stage specificity, in their respective hosts [18,19,43]. To date, the miRNA expression profiles of *E. granulosus sensu stricto* (76 known miRNAs, including adults, cysts, and protoscoleces) [44,45], *E. canadensis* (46 known miRNAs, including cysts and protoscoleces) [12,46], and *E. multilocularis* (46 known miRNAs, including cysts) [1] have been reported (Table 1). Mature miRNAs and pre-miRNA sequences in *E. canadensis* (G7) and *E multilocularis* share an average sequence identity of 98.4% and 99.1%, respectively [1]. Furthermore, miRNAs share ≥87% identity between *E. granulosus sensu stricto* and *E. multilocularis* [44]. However, 22 conserved miRNA families (involved in ciliated cells, the gut, and sensory organs) are not detectable in *E. granulosus* [44].

### 2.1. MiRNAs Expressed in Different Developmental Stages of E. granulosus Sensu Stricto

MiR-2, miR-71, and miR-125 have the highest expression levels among the 76 known miRNAs of *E. granulosus sensu stricto* [44,45]. Interestingly, the expression levels of miR-124b* and miR-87* are higher than those of their mature miRNAs, which suggests that they act as effectors during development and derivatives of their corresponding pre-miRNAs produce two different regulatory small RNAs [45]. In addition, miRNAs were found to exhibit tissue- and phase-specific expression [45]. MiR-277, let-7, miR-71, miR-10, miR-2, and miR-9 are specifically expressed in the cysts walls of secondary hydatid cyst and protoscoleces of G1 and G7 genotype, whereas miR-125 is only detected in protoscoleces and pre-microcysts. Additionally, three miRNAs (let-7, miR-71, and miR-2) are expressed at high levels in protoscoleces of metacestodes (cyst walls), which suggests that their expression is developmentally regulated [45]. Gene Ontology (GO) enrichment analysis revealed that the differentially expressed miRNAs in *E. granulosus* and their potential targets may participate in nutrient metabolism and bi-directional development of the nervous system [44].

### 2.2. MiRNA Expression Profiles in E. canadensis

Among *E. canadensis* G7 miRNAs, bantam, miR-281, miR-184, miR-3479, miR-61, and miR-new-3 have been verified to be expressed in cyst walls G7 (CWG7) and protoscoleces G7 (PSG7) [12]. MiR-71 and let-7 are the most abundantly miRNAs expressed, accounting for 50% of total miRNA expression in two stages that are essential for *Echinococcus* survival in the intermediate host. Moreover, miR-4989 is verified as one of the most abounding miRNAs expressed in cyst wall tissues [12]. Fifteen miRNAs are differentially expressed between the cyst walls and protoscolex of *E. canadensis* G7 (miR-219-5p, miR-4989-3p, miR-27-3p, miR-190-5p, and miR-61-3p are downregulated in CWG7; miR-8-3p, miR-133-3p, miR-96-5p, miR-281-3p, miR-124b-3p, miR-7a-5p, miR-124a-3p, miR-7b-5p, miR-153-5p, and miR-125-5p are upregulated in PSG7) and may be integral to the maintenance of stage-specific features [12]. MiR-4989 and miR-277 may share the same seed region, which suggests that they may also share some target genes that are involved in the preservation of germinal layer features [12,24]. MiR-125 is the most highly upregulated miRNA in PS. Each miRNA family of *E. canadensis* had varied quantities of targets of, for example, miR-2 (miR-2b/2c) and miR-71 families [45]. GO enrichment analysis revealed that potential miRNA targets in *E. canadensis* are involved in protein phosphorylation, transmembrane transport, and transcription regulation. Target genes participate in the Wnt signalling pathway, endocytosis, and the mitogen-activated protein kinase (MAPK) signalling pathway, according to the Genomes (KEGG) enrichment and Kyoto Encyclopedia of Genes analysis [23].

### 2.3. MiRNA Expression Patterns in E. multilocularis

Of the 46 mature miRNAs in *E. multilocularis* that were identified by high-throughput sequencing, five miRNAs, let-7, miR-10, bantam, miR-71, and miR-9 were the most highly expressed miRNAs in *E. multilocularis* [1]. MiR-31-3p and miR-31-5p were only detected in *E. multilocularis* datasets and not *E. canadensis* (G7) datasets. Some miRNAs, such as miR-96, miR-190, miR-3479b, and miR-184, are intronic pre-miRNAs. In other cases, uridylation of *Echinococcus* miRNAs is common (e.g., miR-9-5p, miR-3479b-3p, and miR-71), and the isomer-U is abundant in *E. multilocularis* [1]. Kamenetzky et al. (2016) reanalysed the miRNA profile based on *E. multilocularis* genome-wide data while using an algorithm (deep architecture) of self-organizing maps (SOMs) [46], from this three new miRNAs were predicted: emu_miR-new2-3p, emu_miR-new3-5p, and emu-miR-new9-3p. Interestingly, the authors discovered miRNAs in *E. multilocularis* that were not identified with the Megablast algorithm (i.e., miR-3479, miR-307, miR-36, and miR-1992) [48].

### 2.4. Common miRNAs in Echinococcus spp.

In this review, all of the identified miRNAs in *E*. *granulosus sensu stricto*, *E*. *multilocularis* and *E. canadensis* (G7) were obtained from the reported articles [1,12,44,45,46]. Across the three species analysed, some miRNAs were shown to be highly conserved, which suggested further functional conservation. Eighty-seven miRNAs, suggesting highly conserved miRNAs may perform crucial roles in the development and parasitism of *Echinococcus* spp. Among these highly conserved miRNAs; miR-71, bantam, let-7, miR-9, miR-10, miR-7, miR-87, and miR-61 were the most highly expressed miRNAs in *E. multilocularis* and *E. Canadensis* (G7) infection of the intermediate host [1,12,44]. The function of the miRNAs miR-71, let-7, and miR-61 were verified [44,49], and the target of miR-71 was also confirmed (Table 2), in total, the targets and biological functions of fifteen miRNAs in *Echinococcus* were predicted. The miRNA let-7 exhibited a substantially increased expression in protoscoleces and cysts, which might be associated with the bi-directional development capabilities of *E. granulosus* [44]. Moreover, Mortezaei et al. demonstrated that, under benzimidazole exposure in vitro, the expression of *E. granulosus* miRNAs let-7 and miR-61 was significantly affected in the microcyst stage; however, these miRNAs exhibited different alteration patterns in response to albendazole sulfoxide in other developmental stages [49]. In addition, the ubiquitin-conjugating enzyme E2 was identified as the potential target of miR-307, suggesting that miR-307 might be involved in ubiquitin-mediated proteolysis and herpes simplex infection signalling pathways in *Echinococcus* [23].

MiRNAs highly expressed in *Echinococcus*, but not expressed in vertebrate host, may have diverged from their host homologue miRNAs, for instance, bantam, miR-71, and miR-277 [12], can be assessed as candidate targets for diagnostic markers and intervention strategies. A previous study found that nematode exosome-derived miR-71 plays an important role interaction between the host and parasite, as an innate immune regulator [50,51]. A mimic of *Echinococcus*-derived miR-71 did not change the level of IL-10 in mouse RAW264.7 cells to evade host immune surveillance [52,53]. Furthermore, it can also be deduced that miR-277 might be involved in regulating Wnt signalling pathways, which are responsible for the regulation of stem cell pluripotency in *Echinococcus* [23]. The conservation of miRNAs that are involved in *Echinococcus* regulation reflects the complex and sophisticated adaptations, which are necessary for different environments, present within the life cycles of parasitic species.

## 3. Non-Coding RNAs in Intermediate Hosts during Infection with *Echinococcus* spp.

High-throughput sequencing and miRNA microarray analyses identified dysregulated miRNAs present during parasite infection in natural hosts and animal models that were present in relevant cells, tissues, and blood; demonstrating the importance of these miRNAs in host responses to pathogen challenges [19,43,55]. Recent research results have shown that circulating non-coding RNAs, including miRNAs and lncRNAs, can be stably detected in the blood of hosts that were infected with *E. granulosus* and *E. multilocularis* [19,43,56] (Table 3). These stably circulating non-coding RNAs have the potential to provide us with an understanding of their roles in the host-parasite interaction, development, and growth; and, could potentially serve as diagnostic targets and therapeutic candidates.

### 3.1. MiRNAs and lncRNAs in Host Responses to E. granulosus

MiRNA and lncRNA profiles change when hosts are infected with *E. granulosus* (Figure 2). The gut of the intermediate host is integral to the process, as the first effector of host defence against *Echinococcus* spp. Sheep are highly susceptible to cystic echinococcosis as intermediate hosts. NF-κB pathway-responsive miRNAs, which are related to the inflammation process, are expressed in significantly higher proportions in CE-resistant sheep than in non-CE-resistant sheep, specifically miR-27a, miR-542-5p, miR-134-5p, miR-21-3p, miR-26b, and miR-671 [57] (Figure 2). It could therefore be concluded from aforementioned results that the differential expression of miRNAs present in CE-resistant and non-CE-resistant sheep may be key in the response of intestinal tissues to *E. granulosus*. Myeloid-derived suppressor cells (MDSCs), which are a heterogeneous population of myeloid cells, are composed of dendritic cells, granulocytes, and terminally differentiated macrophages; parasitic infection results in aberrant MDSC expansion [58]. MDSCs accumulate to high levels in mouse models [19,59] and they have demonstrated an important function in the down regulation of the immune response of T lymphocytes when infected with *E. granulosus* protoscoleces. Several differentially expressed lncRNAs and mRNAs were identified between the normal mice and splenic monocytic MDSCs of *E. granulosus* protoscoleces-infected mice [19] (Figure 2). KEGG pathway enrichment analysis suggests that the lncRNAs co-expressed with mRNAs are mainly primarily involved in regulating the vascular endothelial growth factor (VEGF) signalling pathway, the leishmaniasis, *Salmonella* infection, and actin cytoskeleton [19]. The results showed that the aforementioned transcription factors are known to regulate lncRNA production, several of the most likely transcription factors (PGR, IL6, YY1, and FOSL1) for those lncRNAs were predicted by lncRNA-target-transcription factor network analysis [19]. These transcription factors mainly regulate the lncRNAs FR049933, FR291292, FR110455, and FR400826 and they participate in the MAPK and VEGF signalling pathways that are involved in MDSC function [19]. Specifically, the retinoblastoma gene *Rb1*, the expression of which is associated with abnormal M-MDSC differentiation, and was *cis*-regulated by the lncRNA NONMMUT021591 [19]. Such results show that lncRNAs participate in the immune regulation of the intermediate host, mice, in their defence against *E. granulosus,* and might be useful as specific biomarkers for CE.

### 3.2. Mouse miRNAs Dysregulated during Infection with E. multilocularis

The miRNA expression levels of mice were found to significantly in the sera and livers from mice in different stages of *E. multilocularis* infection [43,60] (Figure 2). Mmu-miR-146a-5p, mmu-miR-107-3p, mmu-miR-103-3p, and mmu-miR-21a-3p were found to be significantly upregulated after four weeks of infection. Furthermore, the expression of mmu-miR-339-5p was significantly upregulated at four weeks post-infection, but did not differ from baseline at eight or 12 weeks post-infection. In contrast, mmu-miR-222-3p was found to be significantly downregulated throughout the process of infection. The infectious stage of *E. multilocularis* can be estimated by the expression levels of these miRNAs. The GO terms enriched in potential miRNA targets are involved in the metabolism, signal transduction, immune response, and gene expression regulation [43]. Among *E. multilocularis*-derived circulating miRNAs, only emu-miR-10, emu-miR-227, and emu-miR-71 have been verified [43]. These *E. multilocularis*-derived circulating miRNAs may be used as potential diagnostic markers in intermediate hosts. The levels of three miRNAs (mmu-miR-378a-3p, mmu-miR-101b-3p, and mmu-miR-192-5p) were significantly decreased 90 days post-inoculation when compared to 30 days post-inoculation in mouse livers [60]. These results can inform further studies of the role of host miRNAs during *E. multilocularis* infection.

### 3.3. Common miRNA Families in the Host Model during Infection with Echinococcus spp.

All of the identified miRNAs in the sheep gut and in mouse macrophages, livers, and sera were collected from published literature [43,57,60]. The confirmed miRNA families were then classified according to their annotations in these articles [43,57,60]. Twenty-two common miRNA families were distinguished in intermediate hosts (including sheep and mice) during infection with *Echinococcus* spp. (Table 3), all of which were upregulated in the sheep gut and showed differential expression levels in mouse macrophages, livers, and sera. MiRNAs (e.g., miR-181 [61], miR-30 [62], miR-365 [63], miR-378 [64], miR-449 [65], miR-99 [66], miR-130 [67], and miR-16 [68]) have multiple target genes, including mRNAs, lncRNAs, and circRNAs; this may be key in determining the common miRNAs involved in the hosts response to infection with *Echinococcus* spp., which exhibit different expression levels, functions, and targets in host sheep and mouse models. Many miRNAs, such as miR-181 [57], miR-21 [69], and miR-27 [70], were implemented in the regulation of the immune response in intermediate hosts that were infected with *Echinococcus* spp. Further research should focus on the functional mechanisms of these common miRNAs in hosts that were infected with *Echinococcus* spp. and their potential roles in the treatment of echinococcosis.

## 4. MiRNAs Mainly Associated with Immune and Pathological Processes during Host Infection with *Echinococcus* spp.

The functions and mechanisms of several miRNAs, such as miR-19b (*E. granulosus*) [91], miR-71 (*E. multilocularis*) [92], and miRNA-222-3p (*E. multilocularis*) [93], have been identified. These miRNAs can be used as potential diagnostic markers during infection with *Echinococcus* spp. In this section, we describe the mechanisms and potential uses of these miRNAs in the diagnosis and treatment of echinococcosis.

### 4.1. MiR-71 as an Innate Immune Regulator in Echinococcosis

Extensive research of miR-71 has been conduced, in particular concerning *Echinococcus* and echinococcosis, revealing that miR-71 is a conserved miRNA that is widely expressed in parasites. Nematode exosome-derived miR-71 can be internalized by host cells and serve as an innate immune regulator [50], performing a significant role in host-parasite interactions [51]. MiR-71 also functions in *E. multilocularis* protoscolex development, in which it is differentially expressed at various developmental stages and is found at higher levels in protoscoles without hooks than those with hooks [54]. The Nemo-like kinase gene (*nlk*) is the target of miR-71, when miR-71 binds with *nlk*, NLK expression is inhibited [54]. Thus, miR-71 might play an integral part in the development of alveolar *Echinococcus* [92]. These results lay the groundwork for further exploration into new drugs acting through miR-71 and *nlk* to treat alveolar echinococcosis. Alveolar *Echinococcus*-derived miR-71 also participates in regulating the immune process in mouse macrophages [53]. MiR-71 mimic-transfected RAW264.7 cells do not show significantly altered levels of IL-10 when compared with negative control-transfected RAW264.7 cells, which exhibit significantly repressed NO production at 12 h post-treatment [52]. NO is involved in affecting immunosuppressive anti-parasite immune responses and limiting parasite infection, suggesting that it has essential roles in early and chronic infections of *Echinococcus* spp. [94]. Some components and molecules, such as crude parasite extracts, a laminated layer, and 14-3-3 proteins, have been identified as being able to inhibit NO release by macrophages [94]. Host macrophages may take up parasite-derived miR-71 that is released into the host microenvironments, body fluids, serum, and plasma. Thus, miR-71 is involved in the regulation of *Echinococcus* spp. development and function in host macrophages and can be useful for studying host-parasite interactions.

### 4.2. miR-19b as an Effective Treatment Biomarker

A previous study described that miR-19b plays a part in various pathological conditions and diseases, such as fibrogenesis, osteosarcoma, and clear cell renal cell carcinoma [91,95,96,97]. In cystic echinococcosis, peri-cystic fibrosis is accelerated by hydatid cyst fluid [98]. The expression level of miR-19 is downregulated during the progression of hepatic stellate cell (HSC) activation and, similarly, it is significantly decreased in the patients liver tissues with cystic echinococcosis, and this was also found to be the case in a mouse model of liver fibrosis [98]. Furthermore, miR-19b expression was found to be significantly downregulated in fibrotic liver samples when compared to that in neighboring normal liver tissues, interestingly COL1A1 mRNA expression showed significant-negative correlations with the expression of miR-19b [98]. Hydatid cyst fluid significantly promotes the proliferation of LX-2 cells by accelerating the transition from G0/G1 phase to S phase, increasing the mRNA and protein expression levels of COL3A1, TGFβRII, COL1A1, and α-SMA, [98]. MiR-19b overexpression in hydatid cyst fluid-treated LX-2 cells leads to the significant suppression of cell proliferation and decreases in TβRII, COL1A1 and COL3A mRNA, and protein expression levels by blocking signal transmission in the TGF-β pathway delaying, or potentially reversing the progression of fibrosis [98]. Thus, hydatid cyst fluid is involved in the progression of fibrosis via the activation of hepatic stellate cells, and the regulation of miR-19 expression is sectional of the mechanism regulating peri-cystic fibrogenesis in cystic echinococcosis. Previous studies have demonstrated that TGF-β/Smad pathway activation is the consequence of infections by *E. multilocularis* [99,100]; the activation of this pathway impacts host-parasite interactions, such as fibrogenesis, hepatic (and possibly metacestode) cell proliferation, and immune tolerance mechanisms [98]. These results suggest that *E. granulosus* can promote fibrosis and restrain liver miR-19 expression by increasing TβRII expression, extracellular matrix production, and activating hepatic stellate cells [98]. Furthermore, these results provide new evidence supporting the involvement of miRNAs in regulating fibrosis in infectious diseases. The overexpression of miR-19 in the liver might be an effective treatment biomarker in intermediate hosts that were infected with *E. granulosus*.

### 4.3. miR-222-3p Modulates Macrophage Immunity

Studies have reported that miR-222-3p is implicated in the regulation of vascular physiology and many malignant inflammatory diseases [93]. During infection, *E. multilocularis* has been shown to dysregulate the expression of miRNAs in the liver and serum of infected mice [43,60]. For example, mouse miR-222-3p tends to be downregulated and significantly decreased at two months and three months post-infection, respectively, in the spleens of infected mice as compared with control mice [43]. Furthermore, crude *E. multilocularis* antigens significantly inhibit miR-222-3p expression [6]. Macrophages that were transfected with miR-222-3p inhibitors have been shown to moderately decrease NO secretion, relative to control macrophages. Analysis of transfected cells revealed four key genes implicated in the LPS/TLR4 signalling pathway were found to be significantly down- or unregulated; of which TICAM2, TLR4, and CD14 were upregulated, while AP1 was downregulated [6]. Therefore, miR-222-3p downregulation can modulate macrophage immune functions by regulating NO secretion and the LPS/TLR4 signalling pathway, which potentially contributes to the pathogenesis of alveolar echinococcosis. Thus, miR-222-3p downregulation might be useful as an auxiliary diagnostic marker for alveolar echinococcosis.

## 5. *Echinococcus* miRNA-Related Databases

Various datasets and software programs have been utilized for predicting and analyzing miRNAs in multiple species. Numerous *Echinococcus* miRNAs have been identified via various transcriptomic and modern computational approaches. The application of these databases and software programs could effectively accelerate the exploration of *Echinococcus* miRNA functions and mechanisms (Table 4). The Wellcome Sanger Institute and Sequence Read Archive provide raw genome and miRNA sequencing data, respectively (Chinese Human Genome Centre at Shanghai, and Trust Sanger Institute) [12,14]. *Echinococcus* miRNAs can be authenticated with miRBase [101] and Rfam [102]. Furthermore, miRanda [103], RNA22 [104], RNAhybrid [105], and TarBase v6.0 [106] have been used for predicting and evaluating the targets of miRNAs.

## 6. Techniques and Methods Used in miRNA Studies in *Echinococcus* and Echinococcosis

MiRNA studies in *Echinococcus* and echinococcosis have primarily explored miRNA identification, functions, and mechanisms. Numerous sequencing, bioinformatic analyses, and experimental verification techniques have been utilized to precisely characterize miRNA expression profiles and function mechanisms.

### 6.1. MiRNA Identification

Sanger sequencing [45], modern sequencing techniques (such as RNA sequencing (RNA-seq)) [1,12,44], and bioinformatic analysis approaches (e.g., self-organizing map analysis) [46] have been used to identify miRNAs in *Echinococcus* and echinococcosis. Among these methods, modern techniques (e.g., the Illumina Genome Analyzer II for small RNA sequencing) provide a high-throughput approach for the large-scale detection of miRNA expression in *Echinococcus* and echinococcosis. The validity of the sequencing results and miRNA expression levels can be subsequently verified via Northern blotting [45] and qRT-PCR [12,60]. Bioinformatic analysis, an accurate and convenient approach, is an effective tool that can be used for further verification of new and known miRNAs [46]. For example, a novel deep architecture of SOMs was used to predict novel miRNAs that are based on the complete genome of *E. multilocularis* without the need for RNA-seq data or target analysis for prediction. In theory, this methodology can be easily adapted and applied to any draft genome [46]. However, in practice, these techniques have some limitations. Sanger sequencing is a first-generation sequencing technique, however it is unsuitable for large-scale sequencing. QRT-PCR demands more advanced experimental techniques, but it has the discommodity of high cost and low throughput. The development of high-throughput sequencing technology and the continual expansion of genome libraries has ushered in recent bioinformatic approaches for miRNA discovery (such as the Megablast algorithm and SOM analysis). Despite these advancements, bioinformatic analyses based on big data can yield false positive results, thus further advancements needs to be made to improve the accuracy of experimental tests and provide the necessary verification for the results of bioinformatic analyses.

### 6.2. Verification of the Functions and Exploration of the Mechanisms

Although many *Echinococcus* and echinococcosis-related miRNAs have been identified, the functions and mechanisms of action of most are unclear. Loss-of-function (LOF) and gain-of-function (GOF) studies are usually employed to examine gene functions [110]. The in vivo study of miRNA functions in *Echinococcus* proves difficult under the current achievable experimental conditions. Nonetheless, specific miRNAs from *Echinococcus* hosts, including miRNAs from *Echinococcus* itself, have been identified. LOF studies of echinococcosis-related miRNAs can be carried out in host cells, such as hepatic stellate cells [98] and mouse macrophages [53], with inhibitors or siRNAs [6]. In vitro GOF studies of echinococcosis-related miRNAs in host cells have been performed via the transfection of gene mimics [98]. Additionally, crude *Echinococcus* antigens have been used to treat host cells to study miRNAs and changes in the expression levels of their targets [6].

## 7. Conclusions

To date, a series of miRNAs have been identified in *Echinococcus* spp., but the function and mechanism of most have not been validated. More advanced methods need to be applied to identify effective miRNAs and their functions to deeply understand parasite physiology and to screen for useful diagnostic and treatment targets. As one of the most important hosts of *Echinococcus* spp. and due to the inadvertent nature of infection to humans, more sensitive and discriminatory diagnostic indicators are needed for human echinococcosis in the early stage of infection. Although recent studies have concentrated on miRNAs in *Echinococcus* spp. metacestodes, more research should be conducted on non-coding RNAs in adult cestodes and definitive hosts. Several reported functional miRNAs, such as miR-71, miR-19b, and miR-222-3p, have potential applications in the study of host-parasite interactions and as treatment targets in echinococcosis, therefore they should receive the increased attention. However, despite their potential, clinical application of these functional miRNAs is distant. Moreover, the function and mechanism of action of many identified miRNAs remain unknown. Future research should devote ample attention to screening for miRNA-based early diagnostic markers and treatment targets for echinococcosis in hosts.

In addition to miRNAs, other non-coding RNAs, such as lncRNAs and circRNAs, may play regulatory roles in *Echinococcus* spp. and echinococcosis. Recently, the competing endogenous effect has contributed to our understanding of miRNA regulatory mechanisms at the post-transcriptional level. Protein-coding RNAs, lncRNAs, pseudogenes, and circRNAs act as miRNA sponges. Furthermore, these miRNA sponges interact with each other through shared miRNAs and participate in crosstalk to develop miRNA-mediated interactions or miRNA sponge interaction networks. Therefore, identifying lncRNAs and circRNAs in *Echinococcus* spp. and echinococcosis is necessary for providing new targets for potential treatment and diagnosis.

## Figures and Tables

**Figure 1 ijms-21-00730-f001:**
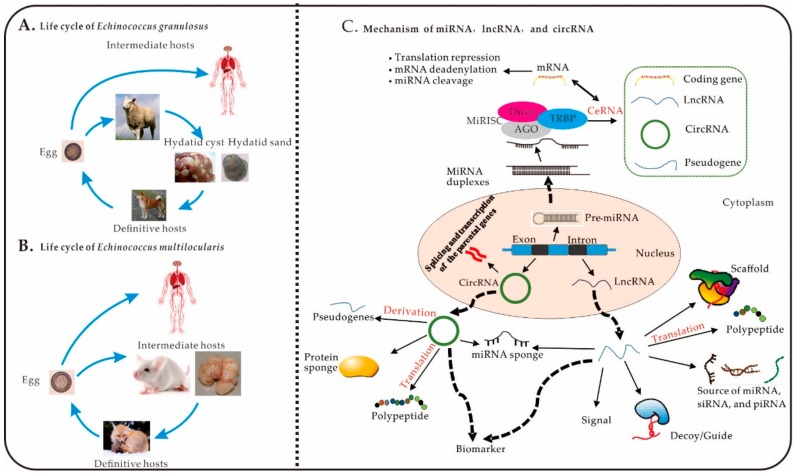
The life cycle of *Echinococcus* spp. and mechanisms of miRNAs, lncRNAs, and circRNAs. (**A**), Life cycle of *E. granulosus*. (**B**), Life cycle of *E. multilocularis.* (**C**) Mechanisms of miRNAs, lncRNAs, and circRNAs: (1) **MiRNA**: pre-miRNA is transcribed from genomic DNA with a double-stranded RNA hairpin and a stem-loop structure in the nucleus [38,39]. The mature miRNA can be loaded into the RNA-induced silencing complex (RISC), resulting of miRNA, Dicer, the RNA-binding protein Argonaute (AGO), and the adaptor protein TAR-RNA-binding protein (TRBP) [40]. The complementary sequences in the untranslated regions of lncRNAs, mRNAs, circRNAs, and pseudogenes can competitively bind to miRNAs, which results in translational repression and degradation of mRNA and miRNA cleavage. (2) **CircRNA**: circRNAs are spliced and transcribed from genomic DNA and transported to the cytoplasm to perform numerous biological functions. One example of such functions is the translation of derived pseudogenes, sponge proteins and miRNAs, into polypeptides [41]. (3) **LncRNA**: lncRNAs play important biological roles in the cytoplasm; including acting as signalling molecules, decoys, guides, and scaffolds; being translated into polypeptides, and serving as sources of small interfering RNAs (siRNAs), miRNAs, and Piwi-interacting RNAs (piRNAs) [42].

**Figure 2 ijms-21-00730-f002:**
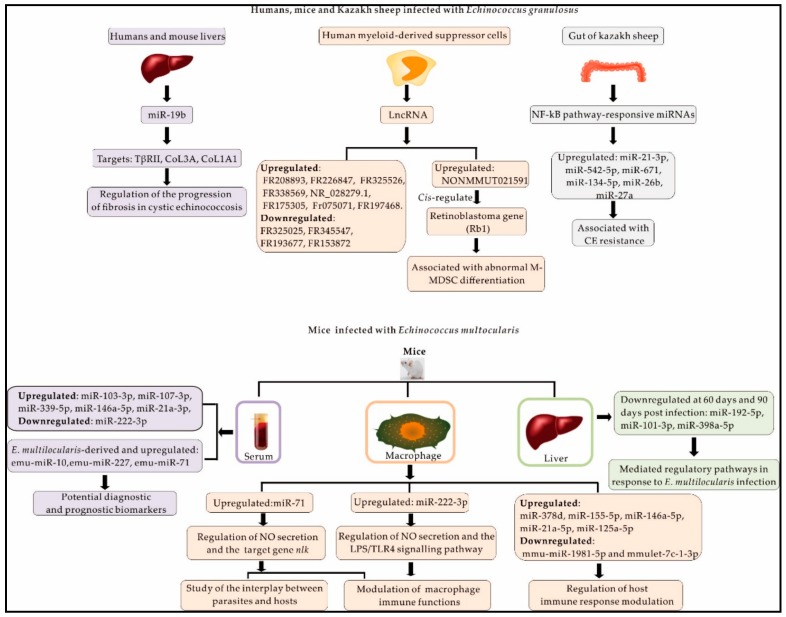
Non-coding RNAs involved in the intermediate host response to *Echinococcus* spp.

**Table 1 ijms-21-00730-t001:** Widespread Expression of microRNAs (miRNAs) in *Echinococcus* spp. According to Transcriptome Analysis.

Species	Tissue	Source	Genotype	Clean Reads	Identified miRNAs	Technology	Country	Ref.
*E. granulosus*	Protoscoleces	Porcine liver hydatid cysts and bovine lung hydatid cysts	G1	182	34 conserved and four new candidate miRNAs	Sanger	Argentina and Uruguay	[45]
	Protoscoleces	Pig	G7		miR-125, miR-2, miR-71, miR-9, miR-10, let-7 and miR-277			
	Protoscoleces	Sheep	G1		miR-125, miR-2, miR-71, miR-9, miR-10, let-7 and miR-277			
	Germinal layer of secondary cysts	Murine	G1		miR-2, miR-71, miR-9, miR-10, let-7 and miR-277			
	Pre-microcysts	Porcine liver and bovine lung	G7		miR-125, miR-2			
	Microcysts	Porcine liver and bovine lung	G7		miR-71			
	Protoscoleces	Sheep liver	__	21,708,040	109 known and 260 novel miRNAs	Illumina HiSeq^TM^ 250	China	[47]
	Adult	Sheep liver hydatid cysts	__	10,069,724	46 known miRNAs, 92 novel mature miRNAs	Illumina Genome Analyzer II	China	[44]
	Protoscoleces	Sheep liver		11,775,532	45 known miRNAs, 91 novel mature miRNAs			
	Cyst membrane	Dog		8,025,262	45 known miRNAs, 103 novel mature miRNAs			
	Protoscoleces	Naturally infected sheep liver	G1	1,642,112/1,956,161 (two biological replicates)	36 miRNAs	Illumina Genome Analyzer II	China	[12]
*E. canadensis*	Protoscoleces	Naturally infected swine liver	G7	4,065,356/1,882,945 (two biological replicates)	35 miRNAs	Illumina Genome Analyzer II	China	[12]
	Cyst walls			2,487,372/2,117,367 (two biological replicates)	35 miRNAs			
	Cyst walls	Swine liver		16,431,381/16,364,826 (two biological replicates)	42 miRNAs			
*E. multilocularis*	Metacestodes	Female CF1 mice (6–8 weeks old)	__	24,703,158/20,396,074 (two biological replicates)	37 miRNAs	High-throughput small RNA sequencing	Argentina	[1]
	Metacestodes	Naturally infected crab-eating macaque (*Macaca fascicularis)* liver	__	The complete *E. multilocularis* genome (https://www.genedb.org/#/species/Emultilocularis)	24 miRNAs	Self-organizing map analysis	Northern Germany	[46]

Note: “—” means that the genotype was not mentioned in the relevant references.

**Table 2 ijms-21-00730-t002:** Common miRNAs in *Echinococcus* spp.

MiRNA	Description of Target Genes	Biological Function	
MiR-71	Nemo-like kinase	Involved in protoscolex development and regulates host macrophage functions in *Echinococcus*	[52,54]
Let-7	Unknown	May be associated with the capability of *E. granulosus* for bi-directional development, can be significantly affected in the microcyst stage of *E. granulosus* and can exhibit different changes in expression in response to albendazole sulfoxide	[44,49]
MiR-61	Unknown	Significantly affected in the microcyst stage of *E. granulosus* and exhibits different changes in expression in response to albendazole sulfoxide	[49]
MiR-10	MAPKs; ECANG7_04447; ECANG7_01705; ECANG7_09658	May be involved in regulating the MAPK and Wnt signalling pathways in *Echinococcus*	[23]
MiR-124	ECANG7_04102; ECANG7_10164; ECANG7_00514; ECANG7_02390; ECANG7_01054	May regulate development, host-parasite interactions, and stem cell pluripotency; related to the MAPK and TGF-beta signalling pathways in *Echinococcus*	[23]
MiR-184	ECANG7_02390; ECANG7_09002 (casein kinase II); ECANG7_05735 (phosphatidylinositol phospholipase C gene); ECANG7_00867 (calcium/calmodulin-dependent protein kinase gene)	May act in a regulatory loop in miRNA biogenesis in *Echinococcus*	[23]
MiR-277	ECANG7_01278; ECANG7_02522	May be involved in the regulation of Wnt signalling pathways regulating the pluripotency of stem cells in *Echinococcus*	[23]
MiR-281	ECANG7_04919 and ECANG7_00818 (glypicas); Nos-1	Potentially involved in the developmental morphogenesis of *Echinococcus.*	[23]
MiR-2	ECANG7_10172; ECANG7_02390; Nos-1; ECANG7_02601; ECANG7_05326	Potentially involved in ubiquitin-mediated proteolysis and herpes simplex infection signalling pathways, TAFH/NHR1 transcription initiation, and segmentation in *Echinococcus*	[23]
MiR-307	Ubiquitin-conjugating enzyme, E2	May be involved in the ubiquitin-mediated proteolysis and herpes simplex infection signalling pathways in *Echinococcus*	[23]
MiR-7	ECANG7_04919 and ECANG7_00818; ECANG7_03238	Potentially involved in the developmental morphogenesis of *Echinococcus*	[23]
MiR-9	ECANG7_02182; ECANG7_05944	Bromodomain-containing protein is an orthologue of *Caenorhabditis elegans* lin-49, which is involved in nematode larval development	[23]
MiR-96	ECANG7_06901	Unknown, but has a high level of expression, particularly in the *Echinococcus* protoscolex stage.	[23]
MiR-125	ECANG7_01292; ECANG7_01524	May regulate developmental genes in *Echinococcus*	[23]
MiR-36	Unknown	May correspond with the increased regenerative capacity of *E. multilocularis* with respect to that of *E. granulosus* s.l.	[1]
MiR-745	G2:M phase-specific E3 ubiquitin protein ligase	Unknown	[23]
MiR-8	Occludin/RNA polymerase II elongation factor, ELL domain RNA polymerase II elongation factor ELL	Unknown	[23]
MiR-87	Zinc finger, C2H2	Unknown	[23]
Bantam	Ribosomal protein S2	Unknown	[23]

**Table 3 ijms-21-00730-t003:** Common miRNA Families in Intermediate Hosts during Infection with *E. granulosus*, *E. multilocularis,* and *E. canadensis.*

MiRNA	Expression Level				Target	Biological Function	Ref.
	Sheep Gut	Mouse					
		Macrophage	Liver	Serum			
**MiR-1247**	**↑**	-	-	↓	CircUBXN7	Represses cell growth and invasion in human bladder cancer.	[71]
MiR-145	↑	-	↓	-	ZEB2	Increases the apoptosis of activated hepatic stellate cells induced by TRAIL via the NF-κB signalling pathway.	[72]
MiR-181	↑	-	-	↓	Smad7 Hsa_circ_0007385	Influences the differentiation of T helper cells and the activation of macrophages, controls T cell sensitivity to antigens during development	[73]
MiR-18	↑	-	-	↑	Unknown	As the female immunity regulator, miR-18 controls the expression of A20/Tnfaip3 and exacerbating NF-κB-driven inflammation in fibroblast-like synoviocytes of rheumatoid arthritis	[74]
MiR-20	↑	-	↓	-	ATG10	Inhibits autophagy and chondrocyte proliferation by targeting ATG10 through the PI3K/AKT/mTOR signalling pathway.	[75]
MiR-21	↑	↑	-	↑	Different targets engaged in each cell type and at each time point	As the one of the master regulators of innate immunity, miR-21 plays a myriad of roles in various cellular processes via regulating genes involved in signalling pathways, such as p53, FOXO1, TGF-α, apoptosis (PDCD4),P13K/Akt/mTOR, VEGF, and NF-αB	[69]
MiR-22	↑	-	↓	-	CD147, YWHAZ	Inhibits hepatocellular carcinoma cell invasion, migration, and proliferation, miR-22 downregulation predicts poor survival.	[76,77]
MiR-223	↑	-	↓	-	Ras-related protein Rab-1 (Rab1)	May promote apoptosis and suppress cell growth through Rab1-mediated mTOR activation in hepatocellular carcinoma cells. In addition, miR-223 is a biomarker of acute and chronic liver injury	[78,79]
MiR-27	↑	-	↓	↑	GATA3, c-Rel, Smad2, Smad3, lncRNA-CIR	MiR-27 plays the important roles for safeguarding Treg-mediated immunological tolerance	[70,80]
MiR-30	↑	↑	↓	↓	MyD88, lncRNA n379519, lncRNA CNALPTC1	Inhibiting cytokine expression and TLR/MyD88 activation in THP-1 cells during *Mycobacterium tuberculosis* H37Rv infection	[62,81]
MiR-339	↑	↓	-	↑	Skp2	Binding to the 3′-UTR of Skp2 mRNA to inhibit the lung cancer cells proliferation	[82]
MiR-345	↑	-	↑	↓	AKT2	Regulates the cell cycle, apoptosis, and proliferation of acute myeloid leukaemia cells by targeting AKT2	[83]
MiR-365	↑	-	↑	↑	LncRNA MT1DP, Timp3	The lncRNA MT1DP exacerbates cadmium-induced oxidative stress by suppressing the function of Nrf2 acting as ceRNA of miR-365. Then, miR-365 promotes diabetic retinopathy through inhibiting lncRNA Timp3 increasing oxidative stress	[63,84]
MiR-378	↑	↑	↑	-	IRG1, lncGAPLINC	Acts as a prognostic marker and inhibits epithelial-mesenchymal transition in human glioma and acts as a molecular sponge of lncGAPLINC to stimulate gastric cancer cell proliferation	[64,85]
MiR-449	↑	-	↓	-	LncARSR	LncARSR is competitively binding to miR-449 and thereby promoting sunitinib resistance in renal cancer	[65]
MiR-542	↑	-	-	↓	SMAD	After activation of SMAD2/3 phosphorylation and the promotion of mitochondrial dysfunction, upregulated miR-542-3p/5p may reduce muscle atrophy in intensive care of patients	[86]
MiR-877	↑	-	-	↓	Cyclin-dependent kinase 14	Suppresses cell migration, invasion, and growth, and predicts prognosis in hepatocellular carcinoma	[66]
MiR-99	↑	↑	-	-	SMARCA5	Regulates *Mycoplasma gallisepticum* (HS strain) infection by suppressing cell proliferation in chickens	[87]
MiR-124	↑	-	-	↓	LncHOTAIR, lncMALAT1, circMMP9	LncHOTAIR sponged miRNA-124 to promote renal cell carcinoma malignancy through alpha-2,8-sialyltransferase 4. LncRNA MALAT1 acts as a ceRNA to control amadori-glycated albumin-induced MCP-1 expression in retinal microglia through a miRNA-124-dependent mechanism	[88,89,90]
MiR-130	↑	-	-	↓	LncMRPL39	LncMRPL39 inhibits gastric cancer progression and proliferation by directly binding to miR-130	[67]
MiR-16	↑	↓			LncDleu2	LncDleu2 influences the invasion, migration, and proliferation of laryngeal cancer cells via miR-16	[68]

Note: ↑, upregulated; ↓, downregulated; -, not detected.

**Table 4 ijms-21-00730-t004:** *Echinococcus* miRNA-Related Databases.

Name	Website	Description	Reference
Wellcome Sanger institute	https://www.sanger.ac.uk/resources/downloads/helminths/echinococcus-multilocularis.html	Includes *E. multilocularis* and *E. granulosus* genomes	[14]
Sequence Read Archive	https://www.ncbi.nlm.nih.gov/sra/?term=Echinococcus%20miRNA	Includes *Echinococcus* miRNA raw sequencing data obtained by second-generation sequencing	[12,107,108]
miRBase	http://www.mirbase.org	A database of *Echinococcus* miRNA sequences and annotations	[101]
Rfam	http://rfam.xfam.org/search?q=Echinococcus	Includes known *Echinococcus* rRNAs, tRNAs, snRNAs and mRNAs	[102]
miRDeep2	https://www.osc.edu/book/export/html/4389	miRNA prediction	[109]
miRanda	http://www.microrna.org/microrna/home.do	Used to predict the target genes of all mature miRNAs	[103]
RNA22	https://cm.jefferson.edu/rna22/	Used for target predictions for multiple species	[104]
RNAhybrid	https://bibiserv.cebitec.uni-bielefeld.de/rnahybrid/	Used to find the minimum free energy for hybridization of a long and a short RNA for predicting miRNA targets	[105]
TarBase v6.0	http://diana.imis.athena-innovation.gr/DianaTools/index.php?r=tarbase/index	Includes experimentally verified interactions between miRNAs and target genes	[106]

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
