# Peer review of "miRNAs and lncRNAs in *Echinococcus* and Echinococcosis"

_ijms, 2020, doi:10.3390/ijms21030730_

Round 1
Reviewer 1 Report
Summary:
The review presented by Zhi He et al. addresses non-coding RNAs (miRNA and lncRNA) in Echinococcus namely E. multilocularis, E. granulosus sensu stricto, and E. canadensis. The authors give an overview of identified non-coding RNAs, their biological function, and target gens in the three Echinococcus species and in the infected intermediated host alike. Thereby, the authors list identified miRNAs in the three Echinococcus species separately and provide information on common miRNAs. Immune and pathological processes of selected miRNAs were mentioned. In addition, miRNA-related databases and techniques used to investigate miRNAs are listed. The authors pointed out the importance of non-coding RNAs as potential diagnostic makers and therapeutic targets as well as the importance and difficulties to reveal the function of miRNAs and mentioning the lack of studies on the adult stage of the parasite.
This review is generally nicely done and provides a practical up-to-date bibliography on a topic not widely reviewed yet. Nevertheless, I have several concerns to be addresses as stated below.
Major points:
Please separately process multilocularis and E. granulosus sensu lato (especially when mentioning hosts and distribution) in the introduction. In the current form I see a high risk to unintentionally mislead audience less familiar with the differences between the Echinococcus species. The importance of miRNAs for diagnosis and treatment is mentioned several times in the manuscript. I miss the novelty of this statement. The authors may contribute more to the review and increase the novelty of their work by investigating competing statements and limitations to stimulate the debate rather than stating the well-known. The different tissue types and stages (metacestodes, germinal layer, and so on) should be introduced. This is especially of importance to correctly map the miRNAs listed in the tables (table 1).
Following are my point by point comments that might improve the manuscript.
Line 18: remove the double parenthesis
Line 39: In introduction “protoscolices” in table 1 “protoscoleces” (use one type for the whole manuscript)
Line 44: “Thus, an in-depth study of biological characteristics, immune escape, diagnosis and treatment in Echinococcus spp. is important to anti-echinococcosis.» I belief this sentence is out of context. Either remove it or elaborate more.
Line 50: "nt in length,; and” remove double punctuation.
Line 54: I strongly empathize E. granulosus, E. multilocularis, and E. canadensis as the only correct abbreviations instead of Eg, Em, and Ec.
Figure 1: a) The figure “Mechanism of miRNA; lcRNA, and circRNA” may be not enough self-explanatory and the shown illustrations could be more detailed explained in the text body. Check spelling for example lcRNA => lncRNA in the title.
b) The visualization of the life cycle could be improved. There are still text and lines from the life cycle the images were copied from. Also, I suggest to separately visualize the E. multilocularis and E. granulosus sensu lato life cycles.
Line 90 – 91: I disagree that the morphological structures of Echinococcus species are simple.
Line 106: miR-125 was also identified in the pre-microcysts. you mention miR-125 was only identified in protoscoleces.
Line 112: remove the “the” after Among
Line 116-117: PS (protoscoleces) and CW (cyst wall) abbreviation not introduced previous in the text.
Line 139: state the bioinformatic approach by name.
Line 137: “and three new miRNAs were found” change “were found” to were predicted
Line 137 to 140: Please include drawbacks and limitations of the mentioned approach.
Line 144: Pleas include the method used for the blast search and analysis.
Line 151 to 153: Please elaborate more on the connection between the conserved miRNAs and the complex life cycles as well as adaption to different environments.
Line 214: citing software error (miR-181{Jiang, 2018 #202},)
Table 3: citing software error ({Ghorbani, 2017 #74;Jiang, 2018)
Line 280: please be more precise on late period of alveolar echinococcosis.
Line 304 to 305: “As an accurate and convenient approach, bioinformatics analysis is an effective tool that can be used for the further verification of new and known miRNAs[34].” Please elaborate more on the statement that this is an accurate and convenient approach.
Line 324: I prefer to say humans are accidental hosts not intermediate hots.
Author Response
Point-by-point responses to reviewers 1 comments/suggestions
Dear reviewer,
Thank you very much for reviewing our manuscript. We appreciate that the reviewer’s comments. All modifications in the revised manuscript are marked in red color. The followings are our point-by-point responses:
Point 1: Line 18: remove the double parenthesis
Response: The double parenthesis have been removed (page 1, line 16 in the revised manuscript).
Point 2: Line 39: In introduction “protoscolices” in table 1 “protoscoleces” (use one type for the whole manuscript)
Response: “protoscolices” in whole manuscript have been revised as “protoscoleces”.
Point 3: Line 44: “Thus, an in-depth study of biological characteristics, immune escape, diagnosis and treatment in Echinococcus spp. is important to anti-echinococcosis.» I belief this sentence is out of context. Either remove it or elaborate more.
Response: We agree with the reviewer’ suggestion. This sentence was removed from the manuscript.
Point 4: Line 50: "nt in length,; and” remove double punctuation.
Response: The double punctuation have been removed (page 2, line 60 in the revised manuscript).
Point 5: Line 54: I strongly empathize E. granulosus, E. multilocularis, and E. canadensis as the only correct abbreviations instead of Eg, Em, and Ec.
Response: “Eg, Em, and Ec” have been replaced by “E. granulosus, E. multilocularis, and E. canadensis” in the whole manuscript, respectively.
Point 6: Figure 1: a) The figure “Mechanism of miRNA; lcRNA, and circRNA” may be not enough self-explanatory and the shown illustrations could be more detailed explained in the text body. Check spelling for example lcRNA => lncRNA in the title. b) The visualization of the life cycle could be improved. There are still text and lines from the life cycle the images were copied from. Also, I suggest to separately visualize the E. multilocularis and E. granulosus sensu lato life cycles.
Response: Thanks for reviewer¢s the favorable suggestions. We have added the elaborate illustrations for the figure “Mechanism of miRNA; lcRNA, and circRNA” (page 3, line 94-105 in the revised manuscript). The the life cycle of E. multilocularis and E. granulosus sensu lato had been drawed, and described in the introduction part (page 1-2, line 29-49 in the revised manuscript).
Point 7: Line 90 – 91: I disagree that the morphological structures of Echinococcus species are simple.
Response: we agree with the reviewer¢s suggestion. And this sentence had been deleted.
Point 8: Line 106: miR-125 was also identified in the pre-microcysts. you mention miR-125 was only identified in protoscoleces.
Response: Thanks for reviewer¢s remind. This sentence have been revised as “whereas miR-125 is detected only in protoscoleces and pre-microcysts” (page 3, line 118 in the revised manuscript).
Point 9: Line 112: remove the “the” after Among
Response: The “the” after Among have been removed.
Point 10: Line 116-117: PS (protoscoleces) and CW (cyst wall) abbreviation not introduced previous in the text.
Response: The abbreviation of PS (protoscoleces) and CW (cyst wall) were not used, and deleted in the revised manuscript.
Point 11: Line 139: state the bioinformatic approach by name.
Response: “a recent bioinformatics approach” in original manuscript have been modified to the bioinformatic approach name “the Megablast algorithm” (page 5, line 153 in the revised manuscript). Then, “miR-10, let-7, bantam, miR-71 and miR-9 are the top five most highly expressed” have been revised as “the five most highly expressed miRNAs (miR-10, let-7, bantam, miR-71 and miR-9) were identified by high-throughput sequencing[1]” (page 5, line 145-146 in the revised manuscript). Moreover, the concrete approach [“using a deep architecture of self-organizing maps (SOMs)” ] used by Kamenetzky et al (2016) have been added(page 5, line 151 in the revised manuscript).
Point 12: Line 137: “and three new miRNAs were found” change “were found” to were predicted
Response: “were found” have been replaced by “were predicted” (page 5, line 152 in the revised manuscript).
Point 13: Line 137 to 140: Please include drawbacks and limitations of the mentioned approach.
Response: In this part, three approaches were mentioned, including high-throughout sequencing technology, SOMs, and the Megablast algorithm (page 5, line 145-146, line 151, line 153 in the revised manuscript). The drawbacks and limitations of the mentioned approach have been analyzed in part 6.1 (line 337-340 and line 342-346 in the revised manuscript).
Point 14: Line 144: Pleas include the method used for the blast search and analysis.
Response: In “2.4. Common miRNAs in Echinococcus spp.” part, we just reanalyzed and summarized the common miRNAs in Echinococcus spp. without miRNA families analysis, and the common miRNAs have been updated in Table 2. Thus, the method used for the blast search and analysis were not added. Furthermore, the method used for the blast search and analysis had been added in 3.3 part “Common miRNA Families in the Host Model During Infection with Echinococcus spp.”(page 8, line 237-239 in the revised manuscript).
Point 15: Line 151 to 153: Please elaborate more on the connection between the conserved miRNAs and the complex life cycles as well as adaption to different environments.
Response: The connection between the conserved miRNAs and the complex life cycles as well as adaption to different environments have been described in details (page 5, line 162-171; page 7, line 178-182).
Point 16: Line 214: citing software error (miR-181{Jiang, 2018 #202},)
Response: This reference has been revised.
Point 17: Table 3: citing software error ({Ghorbani, 2017 #74;Jiang, 2018)
Response: These references has been revised.
Point 18: Line 280: please be more precise on late period of alveolar echinococcosis.
Response: In this part, we mentioned that “mouse miR-222-3p tends to be downregulated and significantly decreased 2 months and 3 months post-infection in the spleens of infected mice compared with control mice[30]”. However, the concrete stage of alveolar echinococcosis can not be determined by expression level of miR-222-3p according to current studies. Thus, the sentence “Downregulated miR-222-3p might be useful as a potential auxiliary diagnostic marker for the late period of alveolar echinococcosis” was revised as “Downregulated miR-222-3p might be useful as a potential auxiliary diagnostic marker for alveolar echinococcosis” (page 11, line 313-314).
Point 19: Line 304 to 305: “As an accurate and convenient approach, bioinformatics analysis is an effective tool that can be used for the further verification of new and known miRNAs[34].” Please elaborate more on the statement that this is an accurate and convenient approach.
Response: We have elaborated the bioinformatics analysis (page 12, line 342-346).
Point 19: Line 324: I prefer to say humans are accidental hosts not intermediate hots.
Response: We agree with the reviewer¢s viewpoint. “intermediate host” have been revised as “accidental hosts” (page 13, line 362).

Reviewer 2 Report
The authors set out to carry out a review of the current body of knowledge on non-conding RNAs in Echinococcosis. The subject is worth writing about and overall the manuscript has merit. However, changes are needed before it can be considered for publication.
Introduction
- The introduction, and the manuscript in general, presents a common problem when dealing with Echinococcosis. The two parasites are related but cause two completely different diseases. As such, they should be illustrated in separate sections to avoid readership confusion.
When describing CE and AE epidemiology we should consider that E. granulosus is a cosmopolitan pathogen, I would avoid listing places as it's never an exhaustive list. Moreover, E. multilocularis is (luckily) confined to the northern emisphere. Again, readership confusion. The eggs do not "transmigrate" anywhere, the exachant larva does.
There is no definitive evidence of dog contact being a risk factors for CE. For AE, where the definitive host is the fox, this is even more unlikely. The use of the term animal contact is vague and readers could think that contact with intermediate hosts is also a risk factor (many do, and again, no hard evidence of this). The use of the term anti-echinococcosis makes no sense. The sentence in lines 53-56 This sentence makes it look like E. canadensis, Em, and Eg are the hosts, while we are talking about the parasites themselves. In Figure 1, only one parasitic cycle is presented for both AE and CE. This is wrong. No method section is present. As such, we do not know how the authors retrieved papers, what was the inclusion process for papers and bio-markers. Some references seem out of place for a number of reasons. Some of the references are non-indexed (e.g. n. 66). Ref. 37 shows a publication from 1990, and miRNAs were not known then, let alone CE or AE-related miRNA. SOme of the biological functions illustrated in the tables are copy-past of refs titles. In the tables about ncRNAs function a mix of information fromm humans, Echinococcus and other parasites. If the function in Echinococcosis is known, I'd state it, if not I would just go with unknown. I do get that it's nice to have lots of references but it hinders readability in this case. For other miRNAs, hypothesised functions are presented. Cases in which more solid evidence has been found should be distinguished from those where this has not been possible. In lines 207-2012 sentences are confusing: do the authors mean to say that miRNAs target the other non coding RNAs (something which is not shown in Figure 2?) or do they mean to say that all the non-coding molecules reffered here have multiplo targets? Morevoer, I do not really get the difference between paragraph 3. and 3.3 content wise, other than the fact that more details are given in 3.3. I would suggest merging the two. A Figure 3 is referred to in the text but not present at all. In line 249 the authors use the term "various diseases" but provide a vague example afterwards ("infectious diseases") Lines 260-266 The authors should more clearly state that they expect mir-19 levels to go down if inactivation of the metacestode is achieved. Line 329 Why thus? The authors first discuss what has been done in intermediate hosts, then what has been done in definitive hosts, then they use a causative conjunction to discuss what should be done for diagnosis. The concepts by themselves are there, but the way they are connected is confusing
Author Response
Point-by-point responses to reviewers 2 comments/suggestions
Dear reviewer,
Thank you very much for reviewing our manuscript. We appreciate that the reviewer’s comments. All modifications in the revised manuscript are marked in red color. The followings are our point-by-point responses:
Point 1: Introduction-The introduction, and the manuscript in general, presents a common problem when dealing with Echinococcosis. The two parasites are related but cause two completely different diseases. As such, they should be illustrated in separate sections to avoid readership confusion.
Response: Echinococcus granulosus and E. multilocularis have been separately described in introduction part (page 1-2, line 30-49).
Point 2: When describing CE and AE epidemiology we should consider that E. granulosus is a cosmopolitan pathogen, I would avoid listing places as it's never an exhaustive list. Moreover, E. multilocularis is (luckily) confined to the northern emisphere. Again, readership confusion.
Response: Thanks for reviewer¢s good constructive suggestions. The list places had been deleted in the revised manuscript, and those content from reviewer have been rewritten in introduction part.
Point 3: The eggs do not "transmigrate" anywhere, the exachant larva does.
Response: The “egg‘’ have revised as “larva”.
Point 4: There is no definitive evidence of dog contact being a risk factors for CE. For AE, where the definitive host is the fox, this is even more unlikely. The use of the term animal contact is vague and readers could think that contact with intermediate hosts is also a risk factor (many do, and again, no hard evidence of this).
Response: We agree with the reviewer¢s suggestions. Thus, the content of the definitive host have been arranged and described in the revised manuscript and figure 1 (page 1-2, line 30-49).
Point 5: The use of the term anti-echinococcosis makes no sense.
Response: It have been deleted in the revised manuscript.
Point 6: The sentence in lines 53-56 This sentence makes it look like E. canadensis, Em, and Eg are the hosts, while we are talking about the parasites themselves.
Response: To better understand what is being said, this sentence was modified as “MicroRNAs (miRNAs) and lncRNAs are widely expressed in Echinococcus, such as E. granulosus sensu stricto, E. multilocularis and E. canadensis [19, 20].” (page 2, line 63-64).
Point 7: In Figure 1, only one parasitic cycle is presented for both AE and CE. This is wrong. No method section is present. As such, we do not know how the authors retrieved papers, what was the inclusion process for papers and bio-markers.
Response: Thanks for the useful advice from reviewer. We have separately redrew the life history of AE and CE. Furthermore, the detailed history have been presented in the introduction part (page 1-2, line 30-49). Endnote type was not removed in previous manuscript so that the style of manuscript disordered could not retrieved. This question have been resolved in the revised manuscript.
Point 6: Some references seem out of place for a number of reasons. Some of the references are non-indexed (e.g. n. 66). Ref. 37 shows a publication from 1990, and miRNAs were not known then, let alone CE or AE-related miRNA.
Response: The citation error in previous manuscript is our negligence. The right reference have been cited. All references have been checked and corrected in the revised manuscript.
Point 7: Some of the biological functions illustrated in the tables are copy-past of refs titles. In the tables about ncRNAs function a mix of information from humans, Echinococcus and other parasites. If the function in Echinococcosis is known, I'd state it, if not I would just go with unknown. I do get that it's nice to have lots of references but it hinders readability in this case. For other miRNAs, hypothesized functions are presented. Cases in which more solid evidence has been found should be distinguished from those where this has not been possible.
Response: Thanks the reviewer¢s good advice in “Table 2. Common miRNAs in Echinococcus spp.”. According to reviewer¢s suggestions, the function in Echinococcosis have been revised in Table 2 (page 6). And the amount of common miRNAs in Echinococcus spp. was redone the statistical analysis, and described in details (page 5, line 157-158).
Point 8: In lines 207-2012 sentences are confusing: do the authors mean to say that miRNAs target the other non coding RNAs (something which is not shown in Figure 2?) or do they mean to say that all the non-coding molecules referred here have multiplo targets?
Response: This sentence mean to say that miRNAs have multiple target genes. According to the recent studies, miRNA have multiple target genes, including mRNAs, lncRNAs, and circRNAs. The targets of miR-181[61], miR-30[62], miR-365[63], miR-378[64], miR-449[65], miR-99[66], miR-130[67], and miR-16[68] in table 3 in the revised manuscript can prove the point above. Thus, we have been revised it (page 8, line 242-245).
Point 9: Moreover, I do not really get the difference between paragraph 3. and 3.3 content wise, other than the fact that more details are given in 3.3. I would suggest merging the two. A Figure 3 is referred to in the text but not present at all.
Response: Thanks for reviewer¢s suggestions for paragraph 3. and 3.3 content. The paragraph 3.3 content was about the common miRNA families in the host model when infected with Echinococcus spp.. If the paragraph 3. and 3.3 content are merged, the content of paragraph 3. will appear to be too much. In our opinion, this part should be separated from paragraph 3. Furthermore, we have been added the methods how to analysis miRNA families in paragraph 3.3. Moreover, the mistake of figure 3 in this section have been deleted.
Point 10: In line 249 the authors use the term "various diseases" but provide a vague example afterwards ("infectious diseases").
Response: In order to better understand the content of this sentence, the examples of miR-19b have been deleted in previous manuscript. Then it is revised as “A previous study reported that miR-19b plays a role in various disease, such as fibrogenesis, osteosarcoma, and clear cell renal cell carcinoma[67, 92-94]” (page 10, line 279-280).
Point 11: Lines 260-266 The authors should more clearly state that they expect mir-19 levels to go down if inactivation of the metacestode is achieved.
Response: This section have been supplemented in the revised manuscript (page 10, line 288-298).
Point 12: Line 329 Why thus? The authors first discuss what has been done in intermediate hosts, then what has been done in definitive hosts, then they use a causative conjunction to discuss what should be done for diagnosis. The concepts by themselves are there, but the way they are connected is confusing
Response: In order to illustrate the concepts, we have described this section in details (page 12, line 365-370).
Reviewer 3 Report
The paper is well written and organised.
Author Response
Point-by-point responses to reviewers 3 comments/suggestions
Dear reviewer,
Thank you very much for reviewing our manuscript. We appreciate that the reviewer’s comments. All modifications in revised manuscript are marked in red color.
Round 2
Reviewer 1 Report
Thank you for the incorporation of my suggestions. Nevertheless, I still have a few minor points I would like to address.
Line 41: For E. multilocularis I would emphasize more on the fox (fox and dogs).
Line 44: "echinococcosis in the liver results"
I would state "mainly" as few (rare) cases in lung and brain are described for E. multilocularis.
Line 49: "the parasite species ingested"
I would remove "ingested" to avoid misunderstandings.
Line 70: source 24 investigates MicroRNAs in apicomplexan. I miss to see the connection to helminths?
Figure 1: Maybe change the image of the definitive host to a fox for E. multilocularis. I would also separate humans in figure 1A as it is shown in figure 1B as humans are not involved in closing the life cycle.
Line 207: “E. granulosus protoscoleces (E. granulosus-protoscoleces)”
Line 295: The two references [99, 100] using E. multilocularis. Therefore, I suggest writing E. multilocularis instead of Echinococcus spp.
Author Response
Dear editors and reviewers,
We are so sorry for bringing the trouble to editors and reviewers because of language in this manuscript (ijms-639860), and then appreciate reviewer’s feedback and the attached whole iThenticate report.
We had revised the manuscript according to the whole iThenticate report and suggestions, and then modified the duplicate and condensed the language. Some sentences almost the whole paragraph are not matched to duplicated with others published manuscripts (except references and network addresses). All the changes in revised manuscript are highlighted in red.
All authors listed have reviewed the final version of the manuscript and approve it for publication. Since I am not a native English speaker, this manuscript was proofread by an English speaking professional with science background at American Journal Experts Corporation (the verification code 5506-7413-AF01-8946-586P).
We deeply appreciate your consideration of our manuscript, and we look forward to receiving comments from the editors and reviewers.
If you have any queries, please don’t hesitate to contact me at the address below.
Thanks very much for your attention to our manuscript.
Yours sincerely,
Dr. Deying Yang
College of Animal Science and Technology, Sichuan Agricultural University, Chengdu 611130, Sichuan, China; Farm Animal Genetic Resources Exploration and Innovation Key Laboratory of Sichuan Province, Sichuan Agricultural University, Chengdu 611130, Sichuan, China;
Tel: 086-835-2883043
E-mail: Deyingyang@sicau.edu.cn; dnaydy@126.com.
Point-by-point responses to reviewers 1 comments/suggestions
Dear reviewer,
Thank you very much for reviewing our manuscript. We appreciate that the reviewer’s comments. All modifications in the revised manuscript are marked in red color. The followings are our point-by-point responses:
Point 1: Line 41: For E. multilocularis I would emphasize more on the fox (fox and dogs).
Response: We agree with the reviewer’ suggestion. Figure 1 B part has been modified, and dogs have been replaced by fox.
Point 2: Line 44: "echinococcosis in the liver results" I would state "mainly" as few (rare) cases in lung and brain are described for E. multilocularis.
Response: Thank you very much for reviewer very professional comment. This sentence has been revised (page 2, line 53-55 in the revised manuscript).
Point 3: Line 49: "the parasite species ingested". I would remove "ingested" to avoid misunderstandings.
Response: The word "ingested" have been removed (page 2, line 60 in the revised manuscript).
Point 4: Line 70: source 24 investigates MicroRNAs in apicomplexan. I miss to see the connection to helminths?
Response: We are so sorry for our carelessness. The relative reference has been added(page 3, line 99 in the revised manuscript).
Point 5: Figure 1: Maybe change the image of the definitive host to a fox for E. multilocularis. I would also separate humans in figure 1A as it is shown in figure 1B as humans are not involved in closing the life cycle.
Response: Thank for reviewer’s suggestion. Figure 1 A and B have been revised.
Point 6: Line 207: “E. granulosus protoscoleces (E. granulosus-protoscoleces)”
Response: This sentence has been modified (page 8, line 225 in the revised manuscript).
Point 7: Line 295: The two references [99, 100] using E. multilocularis. Therefore, I suggest writing E. multilocularis instead of Echinococcus spp.
Response: “Echinococcus spp.” have been instead of E. multilocularis (page 11, line 318 in the revised manuscript).